# Phased Evaluation of Ontologies with Domain and Ontology Experts

Alex Randles[1,*] and Declan O'Sullivan[1]

[1] ADAPT Centre for Digital Content, School of Computer Science and Statistics, Trinity College, Dublin, Ireland

### Abstract

Ontologies provide an agreed shared terminology which can be used to meaningfully communicate knowledge in a domain. Ontologies are often used in the semantic web to facilitate interoperability of linked data. However, creating high-quality ontologies is a complex and time-consuming task, which should be closely guided. Evaluating an ontology with respect to its application and users is an important step to validate the data modeling abilities. This paper presents an ontology evaluation methodology which was applied during the development of two ontologies which we have previously published. This methodology includes several evaluation methods which were applied during their development to iteratively validate requirements with software tools and expert feedback. The methodology includes evaluations to validate in a structured format the design of ontologies with both domain experts and ontological design experts.

### Keywords
Semantic Web, Ontologies, Ontology Evaluation; Design Methodology.

## 1. Introduction

An ontology is defined as formal specification of shared conceptualizations [1]. Ontologies are commonly applied in a semantic web context to communicate shared terminology (in a machine-readable format) which supports interoperability. Ontologies have a vital role in facilitating collaboration and knowledge sharing within and across communities on the web. Ontologies are commonly used in the semantic web to represent domain knowledge and are typically represented using languages such as OWL and RDF. These languages provide expressive models for defining classes, properties, and axioms, which allow the representation of complex relationships and constraints within a knowledge domain.

Evaluating the ontology with domain experts provides a method to facilitate collaboration in order to ensure that it effectively supports the intend use cases and application in the domain. Evaluating with ontology design experts ensures that the ontology has been correctly designed and implemented according to best practices and standards. The feedback from both type of experts provides a method to validate and refine

*WOP 2024*

*Corresponding author.

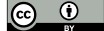 alex.randles@adaptcentre.ie (A. Randles); declan.osullivan@adaptcentre.ie (D. O'Sullivan)

0000-0001-6231-3801 (A. Randles); 0000-0003-1090-3548 (D. O'Sullivan)

the ontology design requirements. This paper discusses the ontology evaluation methodology which we used which involved evaluations that validated ontology design with both domain experts and experts in ontological design. The phased method involves providing both types of experts (domain and ontological design experts) with documentation which follows a standard format. The documentation covers aspects related to the development of the ontology which include definition of requirements, design methodology and previous evaluation methods applied. The experts were asked to review the documentation and complete a questionnaire, which was designed to retrieve key information related to design methodology described in the provided documentation. Furthermore, we outline the existing evaluation methods which were reused from the state of the art and applied prior to expert validation. Moreover, we describe the application of the methodology to two ontologies which we developed, along with the lessons learned as a result.

This paper is structured as follows: Section 2 discusses the ontology design methodology and describes the two ontologies we applied the methodology. Section 3 describes the structure of the evaluation methods and results. Section 4 presents work related to methods used to evaluate ontologies. Section 5 outlines future work and concludes the paper.

## 2. Background on Proposed Ontologies

Previously, we developed two ontologies which were designed to represent information involved in the generation and maintenance of linked data. The **Mapping Quality Improvement Ontology (MQIO)** [2] was designed to represent information related to the quality assessment, refinement and validation of uplift mapping used to create RDF datasets. It extends an existing ontology named the Data Quality Vocabulary (DQV) [3] to represent associated mapping quality metrics and measurements. In addition, it extends PROV Ontology (PROV-O) [4] to represent agents and activities associated with quality improvement of mappings. The **Ontology for Source Change Detection (OSCD)** [5] provides concepts and relationships to represent changes, associated activities and agents which are involved in the maintenance of source data used to generate linked data. The objective of these two ontologies is to support the interoperability and linking of quality-oriented provenance, which is hoped to provide useful information to guide data maintainers when ensuring sufficient data quality. For instance, quality metadata associated with a mapping is represented using MQIO. In addition, changes of the source data of the respective mapping are represented in OSCD. The quality metadata of the mapping provides an indication of quality issues which should be resolved before execution to ensure a high level of data quality in the resulting dataset. Changes which occur in the source data after execution can impact the level of alignment with the mapping, thus impacting the freshness of the data. For instance, the mapping should be re-executed if a significant number of entries have been added to a data reference. The development of the two ontologies followed a similar design methodology, which was strongly inspired by the state of the art. Evaluation with domain and ontological design experts were undertaken in order to discover any issues not found during the development which involved the domain experts alone. The evaluation process resulted in refinement of the ontologies, with the final

versions of MQIO ([www.w3id.org/MQIO](www.w3id.org/MQIO)) and of OSCD ([www.w3id.org/OSCD](www.w3id.org/OSCD)), being published using a persistent identifier.

## 2.1. Ontology Design Methodology

This section discusses the design methodology which was used during the development of MQIO [2] and OSCD [5]. The design methodology and associated results of the evaluations were previously discussed in a PhD thesis [6]. This methodology was inspired by existing reputable methodologies, which include The NeON methodology [7], UPON Lite [8], Ontology 101 development book [9] and LOT methodology [10]. The ontology development methodology involved an iterative process where requirements were defined in the form of natural language (non-functional) and ontology competency questions (functional). The defined requirements were translated to concepts and relationships, then constructed and assessed to ensure no logical inconsistencies were identified within the design. Thus, the development process involved multiple iterations of refinements informed by several streams of information, which were collected from tools, documentation, publication and expert feedback. A high-level summary of the methodology is outlined below.

**1. Identification of aims, objectives, scope:** The design process commenced with the identification of the aims, objectives and scope of the ontology, which were outlined in a document[2]. **Table 1** presents an overview of the sections used to define the ontology requirements specification document. The document outlines requirements, the aims, objectives and scope of the ontology, among other things.

**2. Identify and analyse relevant information:** Identifying relevant information involved completing a state-of-the-art review on the domain in order to retrieve relevant literature. A process commenced by creating a document which stated the inclusion/exclusion criteria which helped to identify whether a publication was relevant. The review of the collected literature helped to retrieve useful insights which were informative during the formalization of competency questions.

**3. Create Use-cases and Competency questions**: Competency questions were created for both ontologies. The questions define the functional requirements of the ontology and were iteratively refined until an accurate representation of the requirements and objectives was conceived. An extract of the final iteration of the competency questions is shown in **Table 2**. Use cases demonstrated that the ontology could be applied to real world use cases.

**4. Identify Concepts and Relationships:** Concepts and relationships were identified through the state of the art review and the researchers' previous experience in the domain. The concepts and relationships were iteratively defined until the information modelling provided by the ontology satisfied each of the competency questions. In addition, concepts and relationships were reused from existing vocabularies as recommended by the methodologies and the W3C recommendation on Data on the Web Best Practices [3].

**5. Progressive iterations:** Steps 2-4 were iteratively repeated until the point when the proposed concepts and relationships provided information which satisfied each requirement defined in the form of a competency question.

---

[2] https://github.com/alex-randles/Ontology-Evaluation/blob/main/MQIO-requirements.pdf

**6. Create Ontology:** It was decided to implement the ontologies in OWL 2 Web Ontology Language[3], which was chosen as it provides sufficient axioms to model the required information and includes an RDF-based representation. Protégé ontology development tool [11] was used to construct the ontology concepts which emerged from the previous steps. However, other ontology development platforms such as the Neon toolkit[4] could be utilized. It is recommended to use a development which includes semantic reasoners to allow logical inconsistencies in the design to be detected and resolved.

**7. Evaluate:** The ontology was evaluated with respect to the ability of the defined concepts and relationships to fulfil each competency question. The usage of a semantic reasoner within Protege ensured logical inconsistencies were identified and resolved. OOPS! Pitfall Scanner [24] was used to detect common ontology design issues. The quality of metadata and documentation was evaluated through presentation within peer reviewed publications. Feedback received from reviewers allowed areas for improvement to be identified and addressed.

**8. Publication:** The documentation for the ontologies was created using WIDOCO [12]. The resulting documentation was updated to include an open and permissive license. Thereafter, the HTML pages were published on the web using a W3id persistent identifier[5].

**Table 1:** Summary of ontology requirements specification document [13]

| Section | Description |
|---|---|
| ***Purpose*** | Outline the rationale for the creation of the ontology. |
| ***Scope*** | Define the areas which are in and out of scope of the intend ontology usage. |
| ***Implementation Language*** | An optional section which outlines the intended language to be used to implement the ontology. |
| ***Intended End-Users*** | An optional section which outlines the target audience of the ontology. People who are expected to benefit from its creation. |
| ***Intended Uses*** | An optional section which outlines proposed use cases of the ontology. |
| ***Ontology Requirements*** | Functional requirements relate to the knowledge and content of the ontology. It is recommended to represent as competency questions. Non-functional requirements relate to the characteristics of the ontology and not the content. |
| ***Glossary of Terms*** | An optional section which defines terminology used in competency questions and answers. |

Ontology Competency questions define design requirements in natural language form. These questions state information which should be provided by the ontology. The fulfilment of the questions is accomplished by providing a concept or relationship which represents the required information. Most questions were inspired by literature reviewed in the state-of-the-art. However, certain questions were defined through application to use cases and feedback from experts. **Table 2** shows an extract of final iteration of competency questions[6]

---

[3] https://www.w3.org/TR/owl2-overview/
[4] http://neon-toolkit.org/wiki/Main_Page.html
[5] https://w3id.org/
[6] https://github.com/alex-randles/Ontology-Evaluation/blob/main/MQIO-competency-questions.pdf

created for MQIO. A description of each concept and relationship used is available[7]. The "Question" column includes the competency questions. The "Relationship" includes the property in the ontology which represent information related to the answer. The "Concept" includes classes in the ontology which represent information related to the answer.

**Table 2**: Extract of MQIO Competency Questions

| Question | Relationship | Concept |
|---|---|---|
| *Who created the mapping?* | mqio:wasCreatedBy | prov:Agent mqio:MappingRefinement |
| *What was the rationale for creating the mapping?* | mqio:hasPurpose | xsd:string |
| *What instruments were utilized to define the mapping?* | mqio:usedTool | xsd:string |

The questions are answered by providing concepts in MQIO which represent the respective information. For instance, the first question states that the ontology should provide concepts and relationships, which can represent who created the declarative uplift mapping being assessed. In addition, to natural language answers, SPARQL query answers to the competency questions are available[8]. Additional information on the graph used to execute the queries can be found in the "Description" section[9] of the ontology documentation.

## 3. Ontology Design Evaluations

This section presents the evaluation methodology which was used to validate the design of the MQIO [2] and OSCD [5]. The evaluation methodology includes experiments which involve different types of experts (domain experts and ontological design experts) assessing the structure and content of the ontologies.

### 3.1. Summary of Evaluation Methods

**Table 3** presents a summary of the evaluation methods applied during the ontology design methodology.

**Table 3:** Summary of methods used in development and improvement of the ontologies

| Evaluation Method | Fulfilment Method |
|---|---|
| *User Experiment (Domain Expert)* | Validating the ontology with respect to their knowledge representation abilities with domain experts. |
| *User Experiment (Ontology Expert)* | Validating the ontology with respect to their development and evaluation approach with ontology design experts. |
| *Fulfilment of Competency Questions* | Competency questions defined in natural language to outline the information stated in the requirements documentation. The questions are then linked with natural language answers that stated ontology |

---

[7] http://www.w3id.org/MQIO#crossreference
[8] https://github.com/alex-randles/Ontology-Evaluation/blob/main/MQIO-competency-sparql-answers.pdf
[9] http://www.w3id.org/MQitsIO#description

| | terms which provided sufficient information. Finally, SPARQL [14] queries were created to retrieve the stated terms from graphs defined using the ontology. |
|---|---|
| *Semantic Reasoners* | Semantic reasoners were utilised in Protégé once the ontology terms were constructed. The reasoners aided in the identification of logical inconsistencies. An iterative approach was adopted to ensure that changes made to resolve inconsistencies did not introduce additional ones. |
| *OOPS! Common Pitfall Detection* | Each iteration of the ontology was input into OOPS! Pitfall scanner [15] to ensure that no inconsistencies were introduced. |
| *Documentation* | Documentation was generated using WIDOCO and published on the web in order to disseminate the design which allows others to review. |
| *Demonstrate application to use-case* | Demonstrate that the ontology can be used to represent knowledge in a proposed use case. |
| *External use-case* | Applying the ontology to external use cases demonstrates the usefulness of the knowledge and shows that they can be applied in real life. |
| *Comparison with State of the art* | Comparison with existing ontologies demonstrates that they contribute to existing knowledge. |
| *Analysis of citations* | Citations of an ontology provides indications of areas where they are being applied. |
| *Dissemination of work* | The ontologies were published using a persistent identifier containing the human readable documentation created using WIDOCO. Thereafter, the identifier was provided to fellow researchers to enable reuse of them. |
| *Peer reviewed publications* | The ontologies were peer reviewed in conferences such as ESWC, ISWC and ISCS. The reviews from each venue were taken into consideration and resulted in refinement of the ontologies. |
| *Reproducibility* | Providing ontologies in multiple formats such as OWL, XML and RDF to enable reuse within various software systems. |

## 3.2. Evaluation Methods involving Experts

This section describes the two evaluation methods which were used to validate the design of the ontologies with domain experts and ontological experts. The questions provided to the experts in both evaluations were defined as a result of a series of discussions between the authors to retrieve the most relevant information.

## 3.3. Domain Expert Evaluation

An initial evaluation[10] was completed to validate the design of OSCD [5] with ten domain experts. MQIO [2] was validated with domain experts using interviews prior to the creation of this method and provided inspiration for the design. Domain experts in this context refers to researchers involved in the subject matter i.e., the creation, maintenance and publication of RDF datasets. Thus, it was decided to recruit post-doctoral researchers with previous experience in the creation and maintenance of RDF datasets. This validation was completed prior to the evaluation with ontology design experts (see Section 3.5) to allow vital feedback to be used to refine the design of ontologies prior to the next evaluation. **Table 4** presents the questions provided to the domain experts. The name of the ontology (<ontology_name>) is represented as a placeholder in the stated questions.

---

[10] https://github.com/alex-randles/Ontology-Evaluation/tree/main/domain-expert-evaluation

**Table 4:** Questions used to validate ontology with domain experts

| # | Question |
|---|----------|
| Q1 | Do you think the <ontology_name> should be altered to include new concepts/relationships? |
| Q2 | Do you think the graph of changes detected generated by the application based on the <ontology_name> (as a vocabulary) could be better organized or presented to the user? |
| Q3 | Any additional comments? |

The questions were designed to ask whether new concepts were needed (Q2) and the presentation of the current concepts (Q2). An open-comment section (Q3) is used to collect diverse feedback not covered by the other questions.

## 3.4. Summary of Evaluation Results from Domain Experts

The responses from the domain experts were reviewed to identify common recommendations to address.

**Adding new properties.** 2 of the experts recommended adding properties to the ontology to represent additional relevant knowledge. A property (:hasPreviousValue) was added to represent the previous value of the changed value which addressed comment, "Maybe by including the previous value for UpdateSourceData". In addition, a property (:wasChangedBy) was added to represent the agent who complete in order to address comment, "provenance data related to who made the changes".

**Reuse of ontologies.** An expert recommended extending PROV-O [4] to represent the changes, however, it was discovered that a more suitable ontology existed to represent changes as specialised events. Thus, it is important to inform experts of the existing ontologies which were considered during the development process.

**Future work.** Several experts provided recommendations for updates that could be completed in future. For instance, an expert stated, "You can even add a free text input where the user can edit in your UI the mapping based on your suggestions.". These suggestions include useful insights from domain experts, which can be used to guide the future direction of the ontology development.

**Affirmation of design.** Positive comments affirmed that the ontology design quality was sufficient in the view of these experts, with comments such as "Seems clear to me" and "It seems to be well presented." supporting this observation.

## 3.5. Ontological Design Expert Evaluation

A subsequent evaluation[11] was completed to validate the structure and content of the resulting MQIO [2] and OSCD [5] with ontological design experts. The evaluation consisted of five participants who have at least ten years of experience in the design and development of semantic web ontologies. Each expert was provided with identical documentation. The evaluation structure involved providing the experts with documentation and artefacts,

---

[11] https://github.com/alex-randles/Ontology-Evaluation/tree/main/ontology-expert-evaluation

which detail MQIO and OSCD. The documentation consisted of the links to documentation and sample graphs, requirements specification, competency questions and a link to the questionnaire. The ontological design experts were asked to complete the questionnaire after reviewing each section of the documentation. The questionnaire included five questions. Four of them asked a specific question (part a) related to the provided documentation and had an associated open comment section (part b), which stated "Can you provide a reason for your response?". **Figure 1** presents an overview of the evaluation structure.

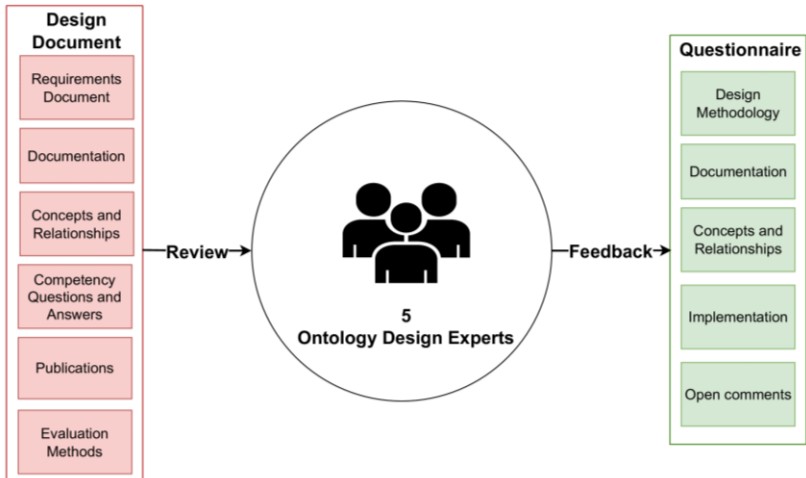

**Figure 1:** Overview of Ontology Expert Evaluation [6]

**Table 5** presents the questions used for evaluating with ontology design experts.

**Table 5:** Questions used to validate ontology with ontology design experts

| # | Question |
|---|---|
| 1a) | In your opinion, does the design of <ontology_name> correctly follow best practices in ontology design? |
| 2a) | Do you suggest any alterations to the design methodology followed by <ontology_name>? |
| 3a) | Do you suggest any alterations to the concepts/relationships in <ontology_name>? |
| 4a) | Do you suggest any alterations to the documentation of <ontology_name>? |
| 5) | Any additional comments? |

The questions were used to test conformance to best practices in the state of the art (1a) and asked for recommendations for the design methodology (2a), contents on ontology (3a) and documentation (4a). In addition, an open comment (5) for other feedback not covered by the initial questions.

### 3.6. Summary of Evaluation Results from Ontological Design Experts

The results of the questionnaire provide quantitative data (number of yes/no) and qualitative data (open comments) to support the observation from the experts. The results for MQIO [2] and OSCD [5] are summarized as follows. Each heading presented relates to a section of the questionnaire.

**State of the Art.** All experts (5/5) stated that both ontologies followed best practices in ontology design practices as recommended in the state of the art. Comments such as "It follows well established methods for developing and iteratively improving the concepts based on application." and "Use of both methodologies and state of the art ontology validation tools" supported these findings.

**Design Methodology.** 4 out of the 5 experts stated that design methodologies followed by both ontologies were sufficient. 1 expert stated that methodologies should include an additional two assessments. These assessments include conformance to the FAIR data principles [16], which are designed to guide data publishers in supporting the reusability of published data assets. An online FAIR validator service was used to test conformance of both ontologies. The Grubers design principles [1] provide comprehensive objective criteria designed to guide the development of ontologies. The five principles are 1) Clarity 2) Coherence 3) Extendibility 4) Minimal encoding bias and 5) Minimal ontological commitment, which encapsulate key criteria's an ontology should satisfy. Both ontologies showed sufficient conformance to these assessments.

**Concepts & Relationships.** 2 out of 5 experts did not provide recommendations for this aspect. The other experts suggested changing the type of a small number of properties from datatype properties (owl:DatatypeProperty) to object properties (owl:ObjectProperty) to capture additional information.

**Documentation.** One expert provided no recommendations to the documentation, while the other experts recommended the following: 1) include sample SHACL[12] constraints designed to assess the quality of graphs expressed in the respective ontologies and include and 2) add an appendix section with sample graphs rather than including them solely as hyperlinks. Thus, it was decided to add sample constraints for validating the quality of respective graphs and an appendix section detailing sample graphs to the documentation of both ontologies.

**Open-Comments.** 4 out of the 5 experts provided no recommendations through the open-comment section, with 1 stating "well done". However, 1 expert recommended replacing the concept dul:Agent[13] with foaf:Agent[14] as it is more prominent for representing agents in the semantic web.

An iterative approach was adopted to assess and address each of the expert's recommendations. The process involved changing concepts and relationships using Protégé [11] ontology development tool. Protégé includes semantic reasoners which were used to ensure no inconsistencies were introduced by the changes to the ontologies. In addition, the updated ontology was assessed using OOPS! Pitfall Scanner [15] to detect common design pitfalls. Finally, the online documentation was updated and regenerated using WIDOCO [12] and republished on the web. Thus, it can be inferred that the final versions of the ontologies have sufficient quality in the view of these experts by addressing each of their recommendations.

---

[12] https://www.w3.org/TR/shacl/
[13] http://www.ontologydesignpatterns.org/ont/dul/DUL.owl#Agent
[14] http://xmlns.com/foaf/0.1/Agent

## 4. Related Work

The state of the art in ontology design evaluation commonly involves the application of tools and competency questions [17]. This section discusses methods commonly used to validate the design requirements of ontologies.

Protégé [11] is a prominent tool designed to facilitate the creation and editing of concepts and relations contained in ontologies. The open-source implementation provides support for ontologies represented in formats, such as RDF, OWL and XML. The tool includes an intuitive GUI to support straightforward interaction by respective end users. Tabs are provided on the interface to allow users to easily navigate the functions provided by the tool. A diverse collection of plugins is available for the tool which are designed to extend the existing functionality. The plugins include reasoning engines, visualisations and support for multiple ontologies to allow merging and management of versions. Semantic reasoners in Protégé facilitate the detection and resolution of logical inconsistencies.

WIDOCO [12] (WIzard for DOCumenting Ontologies) is a tool which generates documentation based on the structure of ontologies input. The documentation contains diagrams, human readable listings of ontology concepts and relationships, and provenance on the version history. The documentation is represented as HTML pages which include hyperlinks to enable straightforward transversing of terms defined in the ontology. In addition, the pages can be extended by integrating customized HTML, such as additional visualisations. WIDOCO was used to generate documentation for MQIO [2] and OSCD [5].

Ontology Competency Questions [17] are used to define the functional requirements of a given ontology. Oftentimes, the questions are defined in a natural language format which states the required information, thus providing validation for the fulfilment of design criteria. The questions can be answered through natural language by stating the ontology concepts and relationships which provide information to answer the questions. In addition, the questions can be answered through SPARQL [14] queries which are executed on the ontology itself or instances within a sample graph. The retrieval of expected information indicates the question has been satisfied by the ontology. The satisfaction of these questions serves as a validation mechanism for the design requirements of the ontologies.

OOPS! (OntOlogy Pitfall Scanner!) [15] is a tool designed to detect common pitfalls in ontologies. The tool was designed to be used by both ontology design novices and experts. The tested pitfalls can be configured by users to detect the most relevant issues for the given context of the ontology. An indicator is associated with each detected pitfall to provide an indication of the level of severity. The indicators are stated as follows: critical (highest severity), important, minor (lowest severity). In addition, OOPS! provides suggestions, which could be used to guide the resolution of identified issues. Addressing critical issues is crucial to maintaining an acceptable level of quality.

Grubers design criteria [1] were proposed to provide guidance during the development process of ontologies. The work proposes five criteria which an ontology should satisfy. The criteria include 1) Clarity: using objective definitions which are documented in natural language 2) Coherence: ensuring that defined axioms are logically consistent 3) Extendibility: providing a foundational model which allows extension for specific use cases 4) Minimal encoding bias: the design of the ontology should not be dependent on a

particular encoding and 5) Minimal ontological commitment: defining the minimum amount of ontology terms to effectively communicate intended knowledge. It states that when following the criteria that trade-offs may be required for different contexts.

While these approaches do not directly involve the engagement of experts, which could reduce the discovery of useful feedback. We propose to utilize the resulting artefacts from them to inform experts of the involved ontology design decisions. Feedback from the information provided to domain and ontology experts can be used to iteratively validate and refine the ontology design, regardless of the knowledge domain being represented.

## 5. Future Work and Conclusion

A phased approach to evaluations of ontology design involving domain experts and ontological design experts, was undertaken and described in this paper. In addition, publication of the final design of both ontologies allows other researchers to apply them in their applications. It is hoped the evaluation methods described in this paper can be reused by others to guide their ontology development process. The ontology evaluation adopted in this research reused a combination of methods from the state of the art, which were designed to assess different characteristics of ontologies. In addition, the methodology included evaluations involving both domain experts and ontological design experts, which provide a method to facilitate collaboration and identify limitations when the ontology is applied in real world use cases. The structured questionnaires which were used during the evaluations can be applied to ontologies regardless of their knowledge domain and provide a method to measure the quality of their design and implementation. In addition, the iterative evaluation methodology allows the design of ontologies to be refined with guidance from software tools and expert feedback. The various streams of feedback are hoped to capture diverse information which can be used to identify and resolve inconsistencies within the ontology structure. Finally, it is hoped the lessons learned provide helpful insights for researchers during the development process.

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
