# OpenReview forum: "Phased Evaluation of Ontologies with Domain and Ontology Experts"
_swsa.semanticweb.org/ISWC/2024/Workshop/WOP — WOP 2024 Oral_

### Official Review · Reviewer_Tb9s · 2024-08-21
**Review 1**

**Rating:** 6
**Confidence:** 3

**Review:**

In this paper, the authors propose a phased approach to evaluations of ontology design, involving both domain experts and ontology engineers. They are provided with documentation in a standard format of the ontology, that experts can review and complete a questionnaire about. This approach has been applied in two use cases, showing how the evaluation method can be reused by others to guide the ontology development process. The material for the evaluation is provided, along with reports by domain experts and ontology engineers.


**Strengths**

S1: Presentation of a method to evaluate ontologies by domain experts and ontology engineers by means of a questionnaire and precise documentation of the ontology to evaluate. This is not only useful, as qualitative evaluations of ontologies still drive the real quality of a work, but makes sure the assessment is more or less standardized. Furthermore, this is in-scope for WOP.

S2: Inclusion of specific requirements related to the demonstration of use-case application, along with external use-case application, analysis of citation and dissemination. Peer-reviewed publications are included as well in the assessment, and are noted as an important way to refine ontologies, explicitly incorporating feedback to the community into the work. Reproducibility is also a valuable point in the assessment, though it is unclear whether open science is implied in this step (see C1).

S2: Potential to use this framework to evaluate AI-generated ontologies (see C2).

S3: Application to two real-world ontologies, which proves reusability and potential to apply this framework in the future.

S4: The ontology requirements specification document is available on Github, as well as competency questions and SPARQL answers.

S5: The paper is clear, and besides the suggestion to restructure the Related Works section, it is easy to follow.

**Weaknesses**

W1: The state of the art could benefit from further detail: While the state of the art is explicitly mentioned (§2 and §2.1), it doesn’t refer to §4 of the paper, about Related work. Therefore, the mentions to the state of the art tend to be a bit unclear as to what is exactly intende by it. Furthermore, it is not entirely clear how exactly the methodology proposed is different from the state of the art.  §4 could be put together with or as a subsection of §2, as it introduces some concepts that might be useful later.

W2: Points unclear in the methodology:
- W2.1: When the methodology is outlined, it is unclear what step is carried by who.
- W2.2: It is unclear who creates the SPARQL queries to retrieve the terms.
- W2.3: Another unclear point is that it is not known whether domain experts with no knowledge representation abilities could be included in this assessment.
- W2.4: With respect to the design methodology, one expert mentioned the need to assess the conformance to FAIR data but eventually this does not reflect in the methodology, why (see W4)?
- W2.5: While it seems SHACL constraints could be useful to assess an ontology, it seems they aren’t present in the evaluation framework.

W3: Missing documentation: while the data is available on Github, the link in footnote 10 for the domain expert evaluation seems broken.

W4: What is meant by reproducibility is unclear, since open science (e.g. are the materials in the paper provided in an open format? see C1) implies reproducibility and not vice versa. This is also important because FAIR principles are mentioned but not made explicit in the methodology.


**Overall Assessment**

The authors provide a useful framework to evaluate ontologies in different domains by both domain experts and ontology engineers, standardizing the process through structured documentation and questionnaires, and demonstrating its practical utility in two real-world cases. This approach could be used in different cases regardless of the domain of the ontology. However, the paper would benefit from clearer integration and explicitation of the methodology with the state of the art and more explicit detailing of roles within the evaluation process, along with making materials more accessible. Despite these areas for improvement, the framework is a promising tool for both manual and automated ontology assessments, with the potential for significant contributions to the field as the usage of AI in ontology generation grows. Therefore, my suggestion is a rating of 6.

**Detailed comments**
C1:  What is meant by reproducibility is unclear. For example, while open formats are mentioned, licenses are not. Furthermore, while PDF is an open format, and is used to present material on Github, it is difficult to reproduce the questionnaire by using it. Therefore, I propose also adding a csv table with the questions of the questionnaire to be provided for future research. Additionally, giving more information about the FAIR validator service could be valuable to the community.

C2: The proposed evaluation framework could be useful to evaluate in a standard way, by experts of different kind, automatically generated ontologies. Now, with LLMs on the rise, there is a need for qualitative evaluation of these systems, especially for what concerns the coverage of the concepts. This could be included in future work, as it is a work that has the potential to be expanded.

---

### Official Review · Reviewer_DggN · 2024-08-26
**Review 2**

**Rating:** 6
**Confidence:** 3

**Review:**

The submission aims to introduce an approach for the evaluation of the engineering and design practices adopted to model two ontologies. The approach involves both domain experts and ontology engineers.

In detail, the authors described the ontology engineering process adopted for developing the two ontologies; the approach essentially relies on existing ontology engineering methodologies (in particular, waterfall methods), with elements of reiteration; particular attention is devoted to CQs and requirements, as well as to the development of documentation. The proposed phased evaluation takes place at the end of the ontology engineering process.
The ontology engineering process is appropriately described, with fundamental details reported within the manuscript and as supplementary materials (via URLs in the footnotes). Wrt this part, a few remarks and questions:
1) At the end of page 4, the authors write, "Most questions were inspired ... feedback from experts". Is it correct to assume that the original ontologies were developed without the direct involvement of domain experts? In the case of affirmative answer(s), the motivations behind this choice could be argued.
2) At the end of page 1, "The feedback from both types of experts provides a method to validate and refine the ontology design requirements". I suggest revising this sentence because the feedback per se does not constitute a methodology.
3) I suggest adopting the term "ontology engineer(s)" rather than "experts in ontological design"; the paper addresses an issue in ontology engineering, so it is plausible to think of ontology engineers as "experts in ontological design" while using a widely-known term.

Moving to the "core" of the contribution - sect. 3 - a few remarks:
4) Page 5, check nouns agreement in Table 3 ("ontology... their").
5) Footnote 9: At this review's time, the URL provided is not working correctly (Not found).
6) Footnote 10 presents the materials for domain experts - which they used for their evaluation. Q1 explicitly talks about "concepts" and "relationships", while Q2 mentions "graphs"; also, it is mentioned that the participants had experience in RDF datasets creation. Therefore, it is evident that the 10 domain experts were somehow keen on knowledge representation. How were the experts selected?
7) 3 out of 4 PDF files (footnote 11) are not accessible at the time of this review.

More in detail, in my opinion, a few points require further elaboration:
a) In 3.4, among the feedback obtained by the domain experts, an engineering-oriented suggestion was presented (i.e., the reuse of a specific existing ontology). The identification of reusable ontologies is not explicitly stated in sect. 2.1. The authors should elaborate on whether the non-inclusion of activities for the identification of reusable ontologies within their custom engineering methodology is voluntary, i.e. if they expect domain/ontology engineers to provide this feedback (reusable candidates) during the evaluation. If it is not voluntary, it should be argued if and how the authors planned to look for existing reusable ontologies - or, if they planned not to reuse any existing model at all.
b) In 3.6 "Design methodology", following one ontology engineer's suggestions, the authors adopted Gruber's five principles as objective criteria for guiding the ontology engineering activities; they concluded that "both ontologies showed sufficient conformance to these assessments". I suggest further elaborating on how these principles were assessed.

In the Related Work section (4), I would have expected to read something about how the proposed phased evaluation differs from existing (and, possibly, similar) approaches; in other words, I believe this section should be completely restructured to underline the novelty elements of the proposed phased approach.

I am concerned about extending the first portion of the evaluation to other ontologies (i.e., to different domains). The authors involved "RDF savvy" domain experts and prepared them questions (Questionnaire footnote 10) adequate to their knowledge and expertise. For any other domain (far from the Semantic Web), the questionnaire should be completely rephrased (in particular, Q2 is very context-dependent). How do the authors plan to make domain expert-based evaluation "universal", so that it can be applied to other domains? Is there a method that can be followed to generate questionnaires, or, in this case, was the questionnaire just one of the possible qualitative methods applicable to eliciting an evaluation from domain experts (for example, for a different domain, unstructured interviews can replace questionnaires)? The sentence "The structured questionnaires ... knowledge domain" should be revised - why are the questionnaires "structured" and not just questionnaires? Were they validated?
Also, the abstract and Introduction explicitly refer to the evaluation of the design (i.e., the ontology engineering process and its output ontology): it is important to state the differences between the proposed approach and existing ones in ontology evaluation (again, section 4 is the best candidate where to host such content).

In general, I think this paper falls in the areas covered by the workshop and succeeds in contributing (although marginally and preliminarily) to a relevant topic. There are a few things to fix, but the contribution is a step in the right direction.

---

### Official Review · Reviewer_syZv · 2024-09-06
**Review for Phased Evaluation**

**Rating:** 6
**Confidence:** 4

**Review:**

This paper provides a structured method for evaluating an ontology based on feedback from ontology experts and domain experts.

While the structure and formalism is much appreciated, I generally find the questions themselves to be difficult to use. A more structured approach might help provide insight into the deficiencies of the evaluated ontology. For example, yes/no questions don't provide insight. On the other hand, qualitative considerations, such as organization or presentation, you might also use a rating scale, instead.

Evaluation is difficult and multifacted, and I wonder if this is too reductionist to be useful to the ontology expert.

However, I think this is a great framework, and will contribute to meaningful discourse at the workshop.